Development of landscape conservation value map of Jeju island, Korea for integrative landscape management and planning using conservation value of landscape typology

Jun Baysok 1
Kim Ilkwon 2
Shin Jihoon 3
http://orcid.org/0000-0001-7077-9273 Kwon Hyuksoo 1 ulmus@nie.re.kr
1 Ecosystem Service Team, National Institute of Ecology , Seocheon-gun, Chungcheongnam-Do , Republic of Korea
2 Sustainable Urban Studies Department, Gwangju Jeonnam Research Institute , Naju-si, Jeollanam-Do , Republic of Korea
3 School of Environmental Horticulture & Landscape Architecture, Dankook University , Cheonan, Chungcheongnam-Do , Republic of Korea
Ghermandi Andrea
Electronic publication date: 2021 Jun 1
Publication date: 2021
Volume: 9
Electronic Location ID: e11449
Received 2020 Nov 12; Accepted 2021 Apr 22
Copyright: © 2021 Jun et al.
Copyright year: 2021
Copyright holder: Jun et al.
License: This is an open access article distributed under the terms of the Creative Commons Attribution License, which permits unrestricted use, distribution, reproduction and adaptation in any medium and for any purpose provided that it is properly attributed. For attribution, the original author(s), title, publication source (PeerJ) and either DOI or URL of the article must be cited.
License URL: https://creativecommons.org/licenses/by/4.0/

Keywords: Landscape typology, Integrated landscape approach, Landscape conservation, Conventional approach, Landscape mangement

Funding: National Institute of Ecology NIE Strategy research-2019-07 This research was funded by National Institute of Ecology, Korea (Project number: NIE-Strategy research-2019-07). The funders had no role in study design, data collection and analysis, decision to publish, or preparation of the manuscript.

==============================
Understanding landscape as a socio-ecological system where systematic interactions occur among diverse ecosystems and human society is necessary for a sustainable landscape and resource management. However, many countries with rapid economic growth, including South Korea, depend on conventional planning and policy decisions to meet increasing demands for the use of specific natural resources. Such resource-oriented planning and policy which neglect considerations for the surrounding landscape can result in conflicts of interest and regulation. We designed a landscape conservation value (LCV) map of Jeju Island, Korea to overcome rising managerial and policy issues with the provision of systematic perspectives of landscape. With a consideration for natural and human-modified characteristics of the landscape, we used landform and land cover data to create fundamental landscape types. Then, the LCV was assigned to each type by a board of landscape experts. Within a study region, we observed relatively high values in registered protected areas and unique landscapes, and areas where high and low values are aligned. The resultant LCV map can identify areas that potentially require an integrated approach to prevent adverse effects caused by a conventional approach.

Introduction

A landscape is comprised of diverse physical products that result from interactions between different phenomena and human behaviors within an ecosystem. Through perception and recognition by humans, a landscape is a single system consisting of natural scientific, social scientific, and anthropological dimensions (Zonneveld, 1989; Zube, Sell & Taylor, 1982). Thus, landscapes require a complex study of its influences, such as the observer’s experience, age, and personality traits, as well as spatial elements, such as the observer’s position and the surrounding environment (Aoki, 2015; Fry, 2001; Kaplan, 1990; Nakamura, 1982; Sayer et al., 2013). Therefore, not all landscapes are managed in the same manner as there are many countries which utilize their national territory as a resource due to rapid development of technologies and economic growth.

It has been commonly agreed that rapid growth of population, economy, dynamic of land use, and land cover changes cause landscape fragmentation (Dewan, Yamaguchi & Rahman, 2012; Li et al., 2010; Opdam et al., 1993; Su et al., 2014). A majority of the landscape has been under-controlled by humans, and during the last century, conventional approaches for maximizing productivity and use of resources induced devastation of diverse landscapes (Denier et al., 2015; Scherr & Wertz, 2019). In many developing and rapidly developed countries with evolving technologies and growing economies, in the management of resources such as timber, water, land for agricultural goods, and settlement, minerals have been secured as their priority to meet their demand. Thus, resource-oriented management has been implemented by pairing specific resources with different ministries or sectors of the government independently, rather than considering a whole landscape which consists of complex interactive ecosystems such as watersheds, pollination networks, habitats, and social flows (Denier et al., 2015). Such sectoral management can possibly result in different policy goals and regulations within the same area, which could induce conflict among diverse stakeholders during utilization and conservation of the land.

Regarding Aichi target 11 announced at the 10th Conference of the Parties to the Convention on Biological Diversity, 193 countries were assigned to expand protected areas by up to 17% of the country’s terrestrial area by 2020. Eventually, 15 out of the top 20 countries in growth rate of Gross Domestic Product (GDP) per capita from 1990 to 2019 fulfilled their goal (UNEP-WCMC, 2020). Despite their fulfillment, 11 out of the 15 rapidly growing countries, such as China, Sri Lanka, South Korea, and eight others, did not achieve neither the global level of Protected Area Management Effectiveness (Hockings, Stolton & Leverington, 2006) nor the connectivity of the protected areas (Saura et al., 2019), whereas slow growing countries, such as the United Kingdom, the Seychelles, the Bahamas, Botswana, Germany, the Congo and others, achieved the rate of either management effectiveness or connectivity far beyond the global level.

To manage a country’s territory with considerations for biodiversity, sustainability, and resilience towards climate change, systemic resource management is critical. In this context, landscape is a socio-ecological system consisting of natural and human-modified ecosystems which is influenced by diverse land use approaches such as ecological, historical, political, economic, and cultural process and activities (Denier et al., 2015). European countries and other developed countries, typically the United Kingdom, have adapted and developed landscape approaches in a way that respects systematic interactions among diverse ecosystems and human society by establishing local autonomy from the 1970s (Golley, 1993; Selman, 2006). The approach evolved from landscape evaluation in the 1970s, which was adapted to preserve the best and leave the rest to support environmental impact assessment by establishing a local autonomy system. Then, to complement landscape evaluation, landscape assessment was introduced in the 1980s to assess landscape based on its quality and quantity. Eventually, landscape character assessment was introduced in the 1990s which was adapted by more than 14 European countries during the European landscape character assessment initiative from 2003 to 2005 (Fairclough & Macinnes, 2003; Wascher, 2005). Landscape character assessment is the most current and critical landscape approach used for policy and decision making as of today, and it involves landscape resource data collection and analysis, and assessment of landscape character based on its aesthetical, psychological, and social values (Fairclough & Macinnes, 2003).

Like other abovementioned developing and rapidly developed countries, Republic of Korea’s GDP per capita quintupled from 1990 to 2019 and ranked as the 18th fastest growing country out of 155 countries (UNEP-WCMC, 2020). In Korea, the rapid growth of economies and technologies began after the Korean war (1950–1953) which mainly depended on quantitative growth of the national power. Therefore, the sectoral approach of managing specific resources remains the same, whereas the quality and interactions within the environment were neglected. With growing concern for the environment, various acts and legislations have been established and revised mainly by the Ministry of Environment, Ministry of Land, Infrastructure and Transport (MLIT), and Ministry of Agriculture, Food and Rural Affairs over the decades.

To enhance regulatory standards for managing landscapes, the MLIT established landscape legislation in 2007. However, the legislation was not compatible with other acts or legislation due to a lack of practical and reasonable standards for a wide range of stakeholders which weakened its regulatory characteristics (Jung, 2018). In 2013, the landscape legislation was revised such that the MLIT minister must establish the master plan of national landscape policy every 5 years, and the mayors of cities with a population over 100,000 have to establish a mandatory regional landscape plan. Since then, the number of landscape projects has grown continuously with 100 projects implemented in 2015, 139 projects in 2016, and 182 projects in 2017, indicating there is a significant amount of landscape-related data accumulating nationwide (Lee, 2018).

Hitherto, different municipalities and ministries have spent significant parts of their budget on landscape resource surveys. However, due to a lack of compatibility of landscape classification and evaluation between ministries’ policies and municipalities’ regulations, landscape resource data could not be utilized for multi-dimensional planning. Like landscape resources, zonings implemented by different ministries and municipalities resulted in conflicts. Managerial conflicts tend to occur where the boundaries of the zoning falls under different jurisdiction, where one is managed for development and the other is managed for strict conservation. Therefore, for more systematic management of fragmented landscapes caused by policy-oriented zoning, there needs to be a fundamental framework compatible with all jurisdictions that can comprehend a whole national territory and prioritize susceptible areas for enhanced management.

In terms of land use planning, landform is proven to have a great impact on natural processes of landscape as it provides distinct visual borders and homogeneous visual shaping throughout their expanse (Martín Duque et al., 2003) and is one of the central variables in the landscape characterization process (Simensen, Halvorsen & Erikstad, 2018). Despite its significant impact on natural processes, landform has been neglected in the majority of zoning processes in Korea, whereas land cover, which includes vegetation, settlement, and human induced factors (Holt-Jensen, 2018), has been independently used for zoning. Considering the fact that humans are characterized as an agent of change in the physical and biological characteristics of the landscape (Zube, 1987), land cover is a substantial variable as it simply represents the current phenomenon of land except it does not show the continuity and naturalness of the land beneath the cover. In other words, landform can be regarded as the physical ‘skeleton’ of the landscape while land cover, consisting of soils and vegetation, mostly provides the ‘flesh’ on the bones (Tudor, 2014). Therefore, to deal with the current zoning issues and increasing complaints by residents related to separate landscape management, a verification of the connection between social and natural phenomena should be carefully considered. To verify the connection between social and natural phenomena, landform data containing continuity and naturalness characteristics can simultaneously contribute to landscape zoning and planning processes with land cover data.

In this study, to enhance the applicability of the landscape management system, we used landform and land cover data simultaneously to construct a landscape type which is utilized as a fundamental framework for prioritization of susceptible areas based on a landscape conservation value (LCV) assigned by landscape experts.

We set three objectives using the LCV map and the fundamental framework. First, to suggest a sustainable method of managing landscape where landscape types with great conservation value and low conservation value align. Second, to suggest a comprehensive method of integrating pre-existing landscape resource data. Third, to introduce the potential impact of using an LCV map to overcome current landscape-related policy issues. Box 1 provides an overview of the key definitions of terms related to landscape.

Box 1 Value criteria of six factors derived from the geomorphological landscape section of the National Natural Environment Survey guidelines used in this research (Cha, Cho & Kim, 2019).

Key Definitions of terms related to landscape used in this research	
Landscape

	
Landscape is comprised of diverse physical products that result from interactions between different phenomena and human behaviors within an ecosystem.

Landscape is a single system perceived and recognized by humans which comprises natural scientific, social scientific, and anthropological dimensions (Zonneveld, 1989; Zube, Sell & Taylor, 1982) and is influenced by diverse land use approaches such as ecological, historical, political, economic, and cultural processes and activities (Denier et al., 2015).

	
Landscape Type

	
Fundamental framework for prioritization of susceptible areas based on a landscape conservation value assigned by landscape experts for enhanced landscape management.

It is the properties of landscapes which consider biophysical dimensions, human-aesthetic dimensions, user participation, and policy dimension (Groom, 2005).

In this study, landscape types are classified based on a combination of 5 different landform and seven different land cover types.

	
Landscape Conservation Value (LCV)

	
LCV is the general idea and goal of prioritizing conservation efforts to landscape types based on six factors described below. More values are added to landscape types with cultural and natural aspects that are likely to become damaged and fragmented when exposed to human induced activities.

	
Meanings and Value criteria of six factors	
Representativeness: Noticeability of characteristics and form of specific landscape. More values are added as the landscape clearly shows the cause of development and formation of the region.

Specificity: A sign of unique natural phenomena and a developing process. More values are added if the landscape has a distinctive formation and structural materials.

Diversity: A degree of how different landforms are scattered within a region. More values are added if there are diverse components of landscape clustered together.

Rarity: A relative scarcity of landforms and landscape components due to regional characteristics. More values are added if the landscape only appears in a specific region.

Irreproducibility: Vulnerability of the landscape or landform when exposed to natural or artificial environmental change and degree of difficulty for restoration. More values are added if the landscape seems to be sensitive to external factors and is associated with previous climate or environmental change.

Educational Value: Possibility of the object to be used for a research and environmental education. More values are added if the landscape provides usability for a research and provides a great accessibility.

	

Materials & Methods

Study site

We selected Jeju Island, Korea as our study site. With an area of 1,864.75 km2, it is the largest island in Korea and is located in the middle latitudes of the north-west Pacific, the geographical midway between Korea, Japan, and China (Fig. 1).

Figure 1 Study site, Jeju Island, Korea, and its protected areas.

Jeju island is globally recognized for its unique landscape formed by volcanic eruptions between 1.8 million and several thousand years ago. The island contains various recognized resources, including the Halla-san National Park (152.94 km2, 8.20% of Jeju Island), Global Geopark (1,847 km2, 99%), Ramsar Wetlands (2.08 km2, 0.11%), the core and buffer zones of the biosphere reserve (287.30 km2, 15.41%), and a natural world heritage site (94.75 km2, 5.08%) which is designated due to its significant scenic and geological value based on outstanding universal value approved by UNESCO (Woo et al., 2013). However, to respect geographical advantages, and the local, historical, and social potentials of the island, the Korean government designated the island as a Special Self-Governing province to establish a Free International City by alleviating administrative regulation and applying an international standard in 2006. Since then, there has been an increasing number of reports insisting that the landscape is exposed to various threats due to over-tourism and developments supporting international exchange. Presently, Jeju Island receives approximately 15 million annual visitors that hope to experience the outstanding ecological and cultural value of the island. The number of visitors is continually increasing and is estimated to reach 45 million in 2035, leading to concerns about landscape damage as a result of over-tourism (Mehmood, Ahmad & Kim, 2019). Furthermore, alleviated regulation for the Free International City is allowing tremendous speculation in real estate by foreign investors which resulted in irregular changes of land cover and deformation of the unique volcanic landform of Jeju Island. Therefore, analysis of the land cover pattern and discontinuity of the landform is a critical approach for conservation of Jeju Island.

Materials

Land cover map

A land cover map is an environmental theme map best reflecting the current status of a ground surface which is an important factor that affects biodiversity, ecosystem health, and the integrity of protected areas (Hansen & DeFries, 2007; Jones et al., 2009; McKinney, 2002). A land cover map consists of both biotic and abiotic features, most often representing human influence on the landscape such as settlement (Holt-Jensen, 2018; Simensen, Halvorsen & Erikstad, 2018) and its vegetation. It represents the natural response to the physical driver controlling the environment, such as landform (Sayre et al., 2014). Thus, land cover is defined as an interactive result of cultural and biophysical phenomena (Mücher, 2009), while proving an extrapolation framework for in-situ data on environmental themes of concern (Jones, 2008). Land cover has been used as an essential feature for a wide range of landscape characterization studies using holistic, biophysical landscape, and a combination of both concepts with different approaches and perspectives (Simensen, Halvorsen & Erikstad, 2018; Walz & Stein, 2014).

Created by the Korean Ministry of Environment, the land cover maps were produced in three different level classes with the purpose of enhancing the efficiency and scientific nature of policy making. The large-class is drawn at a regional scale of 1:50,000 with seven classification items, the middle-class is at scale of 1:25,000 with 22 classification items, and the detailed-class is at scale of 1:5,000 with 41 classification items. In this study, we used the large-class land cover map with seven classification items as the purpose of this study is to prioritize areas for an enhanced empirical approach for more sound landscape management, rather than evaluating landscape with reference to detailed land classification items. The large-class land cover map is similar to the Coordination of Information on the Environment (CORINE) concept used by the Europe Environment Agency (EEA); however, it has been modified to suit the characteristics of a peninsula in East Asia. The Korean Ministry of Environment has been monitoring and recording changes in land cover nationwide since 1998. With technological development, the land cover data have been updated more frequently and improved with greater precision and accuracy. To determine the land cover of Jeju Island, we used revised data from 2016, as shown in Table 1.

Table 1 Similarities between the land cover classification used in this study and the Coordination of Information on the Environment (CORINE) land cover classification.

Corine Land Cover1	Land Cover2	
	1. Developed area	
1. Artificial surfaces	2. Agricultural area	
2. Agricultural areas	3. Forest area	
3. Forest and seminatural areas	4. Grassland	
4. Wetlands	5. Wetland	
5. Water bodies	6. Barren land	
	7. Open water	
Notes:

1 CORINE classification proposed by Bossard, Feranec & Otahel (2000).

2 Land Cover classification proposed by the Korean Ministry of Environment.

Landform map

The nine-unit landscape model (NULM) differentiates soil-landscape units into nine types by determining the spatial distribution and correlation between the topography and soil, based on the movement of energy, water, and substances in the land surface (Conacher & Dalrymple, 1977; Dalrymple, 1968; Park, 2004). Previous studies using the NULM had issues with objectivity as the classified soil-land surfaces were based on qualitative descriptions. To resolve these issues, Park (Park, 2004) presented a soil-land categorization method using a digital elevation model (DEM) and spatial analysis techniques. With this method, Park recategorized the landforms in South Korea according to their topographic characteristics, including interfluve, summit, shoulder, fall face, backslope, footslope, toeslope, and channel. Shoulder, fall face, backslope, footslope, and toeslope denote erosional slopes, steep slopes with a gradient ≥45°, slopes with similar levels of erosion and deposition, depositional slopes, and flat land near waterways, respectively. Summit refers to convex slopes at the tops of mountains and stream refers to waterways, but also appears in regions with little or no distribution of other land surface types (Park, 2004). Landform was also categorized into either mountainous or flat land according to characteristics of flat land and footslopes in the lower parts of slopes (Lee, Jeong & Park, 2015; Park, 2004). Here, landforms were categorized using the upslope contributing area and surface curvature in a grid cell through spatial analysis with NULM. (Park, 2004). Upslope contributing areas, which assess the potential for movement of materials, were calculated according to the following equation:

(1) AS=(1b)∑i=1n⁡pi×Ai

Here, the upslope contributing area (AS) was obtained by calculating the product of the area (Ai) and the potential inflowing water (pi) for each upslope cell (i), calculating the sum of the products across all upslope cells (where n is the number of upslope cells), and then dividing this sum by the width between contour lines, which was estimated to be the same as the cell size (b). Surface curvature was calculated using Eq. (2), which reflects the convexity/concavity and gradient of the surface.

(2) Cs=(∑i=1n⁡(Zi−Zn)/din/n)

Surface curvature (Cs) was determined by first calculating the difference between the elevation of a cell (Zi) and its surrounding cells (Zn), divided by the distance between the cells (din), summing all the numbers of surrounding cells, and then dividing by the total number of cells (n). The AS was obtained using the ‘dynatopmodel’ package (Metcalfe, Beven & Freer, 2018) and the surface curvature was obtained by the ‘spatialECO’ package (Evans, 2020) in R software. Landform was categorized according to the slope and correlation between the calculated upslope contributing area and surface curvature. In this study, DEM was used at a resolution of 30 m to calculate the upslope contributing area and the surface curvature. We first categorized landforms based on the study by Park (2004). Then, we recategorized the landforms according to the first expert survey, which suggested landform classes of: “summit”, shoulder as “shoulder”, backslope and fall face as “slope”, footslope as “mild slope”, toeslope as “flat land”, and stream as “channel” to enable a clearer visual distinction of landforms (Fig. 2).

Figure 2 Landform re-classification based on the Nine-Unit Landscape Model (NULM) (Park, 2004).

Landscape types

Landscape classification that is based on clear standards allows people to broadly accept the diverse properties of landscapes and has been widely recognized to improve communication for policies and research (Brabyn, 2005; Mücher et al., 2010). In this study, landform and land cover, referred to as landscape units, underwent a thematic overlaying process and resulted in 35 comprehensive landscape types of Jeju Island. The comprehensive landscape types provide a brief idea of the properties of landscapes while considering biophysical dimensions, human-aesthetic dimensions, and user participation and policy dimension (Groom, 2005) prior to conducting an expert knowledge-based landscape survey to evaluate LCVs.

Methods

Preparation of Landscape types

In this study, to identify top priority areas for enhanced landscape management as an initial stage, three rounds of FGI (Focus Group Interview), field investigation, and two rounds of expert survey were conducted (Fig. 3).

Figure 3 Landscape conservation value generation process.

The participants of the FGIs were the editorial board of the Korea Landscape Council and each round covered different topics. The first round was used to define limitations and issues related to the current landscape management of Korea and special issues within a research site. The second round was used to select an appropriate approach for landscape management of Korea. Throughout the second round of the FGI, the literature on advanced landscape managements of other countries and its applicability to Korea were reviewed, which eventually resulted in the consensus to form a landscape typology of the research site as a start. The third round was used to select appropriate variables to form landscape types based on spatial data availability, data scale suitability, reliability, data typology availability, regeneration interval, spatial continuity, and noticeability which resulted in choosing land cover and landform.

Once land cover and landform classifications were converted to a raster with a 30 m cell size and overlaid, it was possible to build a matrix with 35 different landscape types and verify locations of every landscape type using ArcGIS 10.5. After identifying the location of different landscape types, we visited the study site, Jeju Island, in preparation for the expert survey and to aid the understanding of the survey respondents. On site, we gathered materials consisting of four representative images for each of the 35 landscape types and descriptions of each class of landform and land cover.

Expert Survey for setting conservation value of each landscape type

To assign conservation value to each landscape type, two rounds of expert knowledge-based survey were conducted with the support of 13 board members consisting of three landscape-related industry workers with PhDs, six academics, and four researchers. Two rounds of survey were designed to prevent outliers, as having board members to evaluate 35 different landscape types at first glance without any criteria allows for great uncertainty. The first survey was designed to fully reflect the expert landscape knowledge of the board with clear key research points, the nature and construction process for classification of each variable, specific explanations for each of the classifications, and the representative images taken during the field investigation. Prior to evaluating the conservation value of each landform and land cover classifications, the board members were instructed to comprehensively consider six factors (representativeness, specificity, diversity, rarity, irreproducibility, and educational value) based on their knowledge and experience, as shown in Box 1.

The six factors were derived from the geomorphological landscape section of the National Natural Environment Survey guidelines (Cha, Cho & Kim, 2019) and are considered as important criteria for a wide range of landscape and environmental value assessment studies (Erikstad et al., 2008; Solecka, 2018). For example, ‘representativeness’ and ‘uniqueness’ have been used as criteria to assess perceptual and aesthetic landscape values (Smith & Theberge, 1986) when evaluating both cultural (Risbøl et al., 2000; Norges Offentlige Utrendninger (NOUs), 1983) and natural heritages (Andersson & Löfgren, 2000; Ratcliffe, 2012; Rudberg & Sundborg, 1975). ‘Diversity’ has been considered when measuring aesthetic values of landscape based on its complexity, land cover contrast, and diversity (Frank et al., 2012; Frank et al., 2013) for both cultural (Risbøl et al., 2000) and natural heritages (Andersson & Löfgren, 2000; Gonggrijp, 1981; Ratcliffe, 2012). ‘Rarity’ is one of the most common criterion when valuing vulnerable areas that are under pressure (Erikstad et al., 2008) in both cultural (Risbøl et al., 2000) and natural heritage contexts (Andersson & Löfgren, 2000; Gonggrijp, 1981; Ratcliffe, 2012; Rudberg & Sundborg, 1975). In Korea, ‘irreproducibility’ has commonly been used to evaluate only geographical and landscape value (Kim, 2009; Seo, 2013). However, Price (1976) has stated that the degree of irreproducibility is associated with visitor satisfaction with the character of the landscape as people tend to value places with an unspoiled state. ‘Educational Value’ has been used in several studies to assess the cultural value of landscape (Brown & Raymond, 2007; Brown & Reed, 2000; Smith & Theberge, 1986) and has been used to measure cultural (Risbøl et al., 2000) and natural heritage (Andersson & Löfgren, 2000; Gonggrijp, 1981).

These factors were used to standardize the perception of the board members as they have been widely used by landscape researchers and managers of Korea (Jeon, Han & Kim, 2013; Ju & Woo, 2019; Kim, 2009; Lee, 2017; Seo, 2013). The board members evaluated the conservation value of seven classifications of land cover and six classifications of landform using a Likert scale-based survey (1: Very low, 3: Standard, 5: Very high). To assign the weights (importance), a pairwise comparison survey was conducted where respondents were asked to select either “Very Important,” “Important,” “Merely Important,” “Somewhat Important,” or “Same” for each type of landform and land cover. Finally, respondents were given the opportunity to provide their opinions on the research methodology and level of understanding and imaging of six landform and seven land cover classifications.

Since an intention of the second round of the FGI was to have the same 13 respondents to evaluate the conservation value of 35 landscape types, the second survey was designed based on the results of first survey. To prepare the second FGI, we examined the results from the first survey and decided to simplify the landform types from six to five. We also rephrased text to improve terminology comprehension. The individual experts’ weights for each variable were applied to their ratings, derived from the first survey. The weights were applied by multiplying the relevant rating by a factor of 1.75 for “Very Important,” 1.5 for “Important,” 1.25 for “Merely Important,” and 1 for “Same.” After applying the individual experts’ weights, the landform and land cover type scores were multiplied and re-calculated on a scale of 1–10. The group’s mean rating for each landscape type was calculated using the same method. In the second survey, images representing each landscape type (Table S3) were provided, and the experts were instructed to refer to their own ratings and the overall ratings in the first survey while considering the six factors.

Mapping landscape conservation

After the final survey results were collected, the mean LCVs were derived for the 35 total combinations of landscape types and were re-calculated on a scale of 1–10. To apply and map the mean LCV, each cell was assigned a matrix score. The borders of the culturally and ecologically important Ramsar sites, national scenic sites, Oreum with approved landscape quality, biosphere reserve, urban natural park areas, natural monument, national park, Gotjawal, and natural world heritage site were overlaid (Fig. 4). The mean LCV and standard deviations were calculated for the cells pertaining to each site, and these values were compared with the mean for the whole of Jeju Island.

Figure 4 Designated areas of Jeju Island.

To prioritize susceptible areas for enhanced management, the spatial variance of LCV among every single cell assigned with a different landscape type was analyzed using focal statistics in ArcGIS 10.5. Thereby, the standard deviation of each cell and those of the surrounding cells was obtained.

Results

Geographical distribution of landscape type areas

Landscape units, comprising land cover and landform, were separately classified, and their proportions within the research site were overviewed prior to the thematic overlaying process. The land cover of Jeju Island was split into seven categories (Fig. 5). ‘Developed Area’, also known as non-arable land (comprising 13,365.6 ha (7.23%) of the total study area), consists of paved infrastructure for transportation, residential area, industrial, commercial, entertainment, and public facilities. ‘Agricultural area’ (65,608.3 ha (35.49%)), consists of rice paddies, fields, vinyl greenhouse, orchards, and other arable land. ‘Forest area’ (64,181.3 ha (34.7%)), consists of broadleaf tree forest, conifer tree forest, and mixed stand forest. ‘Grassland’ (35,097.4 ha (19.0%)), is comprised of natural and artificial pasture. ‘Wetland’ (724 ha (0.39%)) is comprised of inland and coastal wetland. ‘Barren land’ (5,386.7 ha (2.91%)) is comprised of natural and other barren land. ‘Open water’(492.8 ha (0.27%)), is comprised of inland water and sea water.

Figure 5 Large class land cover map of the research site.

The landform map shows that ‘Flat land’ mainly occupied 122,778.1 ha (66.4%) of the study area, surrounding Halla-san mountain the volcanic origin of the island (Fig. 6). ‘Summits’ (6,610 ha (3.57%)) which are randomly scattered in the form of small points delineating volcanic cones created by eruptions. ‘Sloped lands’ (31,534.6 ha (17.07%)) formed around lava cones and are mainly distributed across Halla-san mountain. ‘Shoulder’ (20,715 ha (11.21%)), in the form of strips, originated from the peak of Halla-san mountain. ‘Channel’ (3,129.9 ha (1.69%)), in the form of strips, formed a network in the ‘Sloped lands.’

Figure 6 Landform re-classification of the research site based on the Nine-Unit Landscape.

As a result of the thematic overlaying process, Fig. 7 shows the major landscape types occupying over 5% of the study area. Flat agricultural land occupies the largest area (582.20 km2, 31.52%), followed by flat forest (301.75 km2, 16.33%), flat barren land (267,16 km2, 14.46%), sloped forest (199.77 km2, 10.81%), flat developed land (118.97 km2, 6.44%), and forest over a shoulder (102.56 km2, 5.55%). The remaining 29 landscape types accounted for 14.89% of the study area, and forests formed over a summit, shoulder, slope, and channel occupied nearly 0% of the research site (Table S2).

Figure 7 Proportion of each landscape type within the study site.

As shown in Fig. 8, the technique developed here is a simple and comprehensive approach that enables the identification of representative landscape types and the location of unique or rare landscape types. Based on the landscape type map, each landscape type was investigated by the research team. Table S3 presents images collected for each landscape type.

Figure 8 Landscape type derived from landform and land cover.

Landscape conservation value

Throughout the expert survey, the LCVs of each landscape type were obtained and normalized on a scale of 1 to 10. Landscape types with the highest conservation value were wetland formed over a summit and open water over a summit (10), followed by forest formed over a summit (9.41), wetland within channel (8.57), open water filled in channel (8.39), and forest formed along the channel (8.23) (Table 2, Table S1). The landscape type with the lowest conservation value was given to flat barren land (1), followed by flat developed area (1.43), hilly barren land (1.59), and hilly developed area (1.68). In Table 2, the landscape conservation value trends are presented with the highest in the lower right corner to the lowest scores in the upper left corner.

Table 2 Final Landscape Conservation Value Matrix based on the results of the 2nd Expert Survey with simplified landform classification and changed land cover class order.

Landform
Land Cover	
Flat land	
Slope	
Shoulder	
Channel	
Summit	
Barren land	1.00	1.59	1.93	2.52	3.02	
Developed	1.43	1.68	2.18	2.43	3.45	
Agriculture	2.86	3.27	4.54	4.54	5.79	
Grassland	4.20	4.79	5.96	6.38	7.98	
Forest	5.63	5.96	7.64	8.23	9.41	
Open water	5.88	6.64	8.14	8.39	10.00	
Wetland	6.21	6.46	7.73	8.57	10.00	

Landscape conservation value map

Figure 9 shows the distribution of 35 different landscape types with an application of normalized Landscape Conservation Values obtained throughout the survey. In Fig. 9, high ratings significantly appear (black) within the boundaries of Jeju Island’s famous landmark, such as Halla-san National Park labeled as 3 in Fig. 9, as well as the other protected regions shown in Fig. 8. The oreums labeled as 1 in the figure, a common name for a volcanic cone in Jeju Island (Nam et al., 2019), are visible as small, scattered circular shapes with high scores; Gotjawal, labeled as 5 in the figure, is a uniquely formed forest vegetation found on lava terrain (located at the eastern and western parts of Jeju Island (Kang, Kim & Kim, 2013)). On the other hand, developed areas such as roads and the central town of Jeju city and Seogwipo city appear to have relatively low landscape conservation value.

Figure 9 Landscape Conservation Value map with multiple layers of designated areas (1-Oreum; 2-Urban Natural Park Areas; 3-National Park; 4-Biosphere Reserve; 5-Gotjawal; 6-Scenic Site; 7-World Heritage Site; 8-Ramsar Site; 9-Natural Monument).

The mean score of every cell in each of the designated areas and whole research site was calculated (Table 3). The values for designated areas, especially the National Park, ‘Oreum’, Biosphere Reserve, and Ramsar site were far higher than the mean for the overall study site and showed relatively lower standard deviations compared to the whole research site, as shown in Table 3.

Table 3 Mean landscape conservation value for the protected regions on Jeju Island.

Designation	Area (km2)	Mean LCV1	Std. deviation	
World Natural Heritage Site	2.46	5.40	1.67	
National Park	152.82	6.41	1.18	
Biosphere Reserve	119.32	6.19	1.17	
Urban Natural Park Areas	5.65	5.64	2.14	
National Scenic Site	1.06	5.81	2.18	
Natural Monument	6.90	5.21	1.89	
Ramsar Site	1.88	6.05	0.93	
Gotjawal	3.12	5.78	7.25	
Oreum	81.04	6.34	1.77	
Whole research site	1,847.33	4.35	1.93	
Note:

1 LCV stands for Landscape Conservation Value.

Figure 10 presents areas potentially susceptible to various human activities which are raster cells with the top 20% (1.74) standard deviation. Through comparison with satellite images, it was found that most of these areas are mountain trails, paved roads through the forest and along streams, industrially and commercially developed areas, residential areas, and tourist spots with a significant amount of paved parking lots which are widely exposed to human activities.

Figure 10 Map of susceptible area based on heterogeneity of landscape conservation values.

Discussion

Management of susceptible area

In this study, the LCV map provides the location of landscape type with both high and low conservation value in the form of a raster dataset. Landscape types with high conservation value are wetland, open water, and forests formed over summits or streams. These types are unique or provide wildlife habitat and opportunities for natural leisure experiences, such as fishing, strolling, and hunting. However, landscapes with low conservation value are flat land, hilly barren land, and hilly developed land. These types provide comparably greater opportunities for the development of commercial plots, residential areas, parking lots and roads.

The natural and the human systems develop and expand territories through different processes and the long-term result leaves significant changes in land cover and landforms. However, due to demographics, globalization, and growing economies, humans tend to influence the landscape quicker than nature can adapt (Millspaugh & Thompson, 2009). Furthermore, with evolving technologies, more landscape types are exposed to human development, as evidenced by the increase of residential buildings in wet and rocky areas (LaGro, 1996). Moreover, human impacts such as landscape fragmentation and land ownership parcelization result in land subdivision along various ecosystems, such as small lakes, rivers, and forest areas, which had formerly not been considered for human uses (Millspaugh & Thompson, 2009). Therefore, landscapes with significant conservation value are becoming increasingly susceptible to human developments. Considering that the research site is a globally recognized tourist destination for its natural landscape and the policy is focused on the enhancement of international exchanges, there has been an increasing demand for natural and cultural experiences and opportunity for tourism businesses. Furthermore, as real estate investment of the research site is available for foreigners, a land ownership has been parcelized for development which resulted in conflicts of stakeholders’ interest. Therefore, land outside of designated areas and permitted for development while being adjacent to great natural landscape can be a great target for tourism investment. This will potentially either positively or negatively influence adjacent natural landscape, depending on the landowner’s strategy, policy adjustment, and conservation budget.

Like cracks in the wall of interconnected ecosystem, some of revealed susceptible areas within the designated area may have been well managed with the support of a conservation budget, whereas some privately owned areas are easily exposed to development and domination as the area provides great opportunities for tourism investment. The development, without considering the surrounding landscape, can negatively influence interaction networks of different landscape types ecologically and biophysically, such as connectance or nestedness (Astegiano et al., 2015). Thus, the susceptible areas being proposed can be prioritized as critical management areas, and flexible strategies, such as conservancy zoning, where individual landowners cluster development and leave the large proportion of their land in relatively undisturbed forest cover (Millspaugh & Thompson, 2009), cross boundary management (Harper et al., 2006), and reserve based models, should be considered in order to prevent landscape fragmentation.

Compatibility of LCV map for integrating pre-existing landscape resources

The major function of the LCV is to be compatible for all regions and easily understood. Presently, landscape resource data are continually being collected from diverse perspectives; however, planners and decision makers are limited in their ability to assimilate data that are mutually different in nature. The LCV map is a container for various existing landscape resource data, resulting in its unique ability to produce additional in-depth, high-quality analyses when overlaid with other landscape-related information. The LCV has an objective biophysical basis, and the landscape types are constructed with fundamental variables, meaning that it reflects the fundamental attributes of the landscape. In addition, because land cover and landform apply to the whole surface, excluding the ocean, there are no gaps. Therefore, the LCV map is compatible with any form of landscape-related spatial data (point, linear, or planar) that corresponds to natural, cultural, aesthetic, and perceptual domains. Although the landscape types proposed in this study provide great compatibility with other spatial data and improve communication among different stakeholders, the variation within the landscape types does not reflect an actual phenomenon of a particular landscape area. However, the landscape type system comprises a predictable level of landscape variation which provides a useful reference for the assessment of an individual landscape area’s character and properties. It is also possible to compare an assigned value with the conservation value of landscape types. For example, if the summit and open water landscape character, which had the second highest conservation value, is present within the matrix of the region of interest, this area is likely to be visually sensitive, rare, and ecologically exceptional. However, if the LCV map is overlaid with landscape resource data with high recreational value, therapeutic value, sensory experiences, or views, planners or policymakers could consider that this area has both high landscape and high utility values, or even that its unique characteristics suggest that it could be subject to degradation. As another example, flat barren land and flat development areas are likely to be a target for development due to their low aesthetical and ecological value. However, if the flat and barren land were overlaid with existing landscape resource data, one might discover a world heritage site, or symbolic, spiritual, or educational value. Therefore, even if an element with a certain value is located in a landscape type with low LCV, visual sensitivity, and ecological value, by overlaying highly valued landscape resource data, planners and policymakers could anticipate the likelihood of degradation. Additionally, considering the proximity of flat and development land that has already been developed, specific administrative measures could be proposed that account for factors including distance from conservation targets and population influx.

The landform and land cover variables used in this study are important not only for landscape value, but also for various other analyses. Landform types are used in analyses relating to flood hazard vulnerability (Mihu-Pintilie & Nicu, 2019), ecosystems (Swanson et al., 1988), vegetation patterns (Baartman, Temme & Saco, 2018), and ground water recharge (Lukenbach et al., 2019). Meanwhile, land cover types have been used in analyses regarding ecosystem services (Burkhard et al., 2009; Koschke et al., 2012), accessibility of resources in the construction sector (Ioannidou et al., 2015), and urbanization patterns (Dewan & Yamaguchi, 2009). Therefore, the LCV presented in this study is highly compatible with analytical data from several fields. Being able to link different analytical data will contribute to more valid and logical landscape-related decision making because it forms a base of mutual understanding with different stakeholders.

Potential impact of landscape conservation mapping process on policy issues

As mentioned in the introduction section, there are several landscape approaches, such as landscape characteristic assessment, applied to national policies in many European countries. However, due to rapid growth of the economy and urban sprawl in Korea, there are rising managerial problems related to landscape and conflicts due to non-compatible policies being operated by independent ministries. Currently, there are various types of protected areas, such as national parks, wetland protected areas, and ecological landscape protected areas, designated across the borders of regions countrywide. Hence, depending on regions, the different levels of regulation exist due to different regional planning. The LCV developed in this study offers spatial planners and policy makers the opportunity to analyze a landscape with a more macroscopic and objective perspective, beginning in the initial stages of analysis. The main function of LCV is to help users comprehensively establish landscape types, identify levels of conservation needed, and compare the region of interest with its surroundings. The application of the concept of LCV to any regional planning and national landscape planning can contribute to adjusting differences in the level of regulation and systemic landscape zoning and planning across regions.

Limitations of the study

The LCV map has three major limitations.

(1) The landform resolution is limited to a minimum of 30 m cells; thus, areas under 30 m2 cannot be analyzed. Furthermore, even if the quality of data is improved, the LCV is constructed from fundamental elements of landscapes, and therefore, should be applied to the landscape scale. However, if land cover types were further sub-classified according to use, it would be possible to a develop mutual understanding between stakeholders, and the results could be supplemented by overlaying cognitive-based landscape resource data.

(2) There were no lay participants in the FGI or surveys. Therefore, whether the assigned variables would be deemed relatable by members of the general public could not be verified. However, if the opinions of the general public are reflected by considering different stakeholder groups with a larger number of participants in the future, it could support more valid and logical decision making. The survey process proposed in this study is designed specifically for landscape experts. To engage survey participants from a wide range of stakeholder groups, the survey instructions should be revised with clear and definitive explanations for all 6 factors and value criteria to prevent outliers.

(3) There were issues with the design and instructions of the survey which impacted the experts’ judgment. To obtain clear and distinct landscape types in the LCV, the landscape types were assigned with numbers ranging from 1 to 10. However, criteria for ‘High’, ‘Medium’, and ‘Low’ values were not provided which could be a possible reason for outliers in expert judgment. This problem could be solved by setting the value criteria during FGI, with engagement of all the previous participants, and by a literature review. For further studies that involve a broader range of stakeholders, the value criteria should be reflected in the survey instruction in a user-friendly manner to improve objectivity and integrity of the results.

Conclusions

This study emphasized the current limitations of landscape management in the Republic of Korea, as one of the countries with a rapid growth of economy, and proposed a process of producing an LCV map of Jeju Island as an integrative approach and one of the solutions to deal with current limitations. The limitations are divided landscape management that mainly focuses on utilization of specific resources and the failure of using landscape resource data. To produce an LCV map, a fundamental landscape type map was constructed using land cover and landform data. Land cover and landform are critical variables in this study to produce landscape types representing landscape as a socio-ecological system with a mosaic of natural and human-modified ecosystems. In this context, land cover data are considered an extended interpretation as they reflect human influence on the landscape. However, landform data were used as it has a great impact on natural processes of landscape, providing distinct visual borders and homogeneous visual shaping throughout their expanse. After production of a landscape type map, the LCV for each landscape type was assigned by a group of landscape experts. The purpose of assigning an LCV in this study was to verify areas in need of conservation-oriented management, areas that need to be aware of negative impacts of development, and, most importantly, susceptible areas that require more careful and constant monitoring due to the coexistence of landscape types with high LCV and low LCV.

In addition, our developed process can be applied to any study site. Using fundamental variables to produce landscape type maps and an LCV allows users to extend beyond identifying landscape features, patterns, and susceptible areas for enhanced management in a given region. This instead enables them to compare regions and create macroscopic landscape analyses at the national scale. If additional variables were included to account for diverse present-day ecology, culture, and cognition, we expect that it would present more objective and logical landscape management and policies.

Supplemental Information

Supplemental Information 1 Survey 1: Landscape Conservation Values of Individual land cover and landform types (Out of 5).

Values given to each type of variables based on the first survey with and without weight disposal of 1.25 towards land cover. On a scale of 1 to 5, the landscape conservation values for each land cover type were, in descending order, 4.53 for wetland, 4.46 for open water, 4.15 for forest, 3.84 for grassland, 3.23 for agriculture, 2.69 for developed, and 2.46 for barren land. The values for each landform type were, in descending order, 5 for summit, 4.46 for river/stream, 4.08 for shoulder, 3.69 for mild slope 1, 3.46 for flood plain, and 3.31 for mild slope 2. Landscape experts rated land cover as “somewhat important” (×1.25 weight) compared to landform; after applying a ×1.25 weight to land cover type, the mean score for land cover was 4.53 and the mean score for landform was 4.00

Click here for additional data file.

Supplemental Information 2 Result of 2nd Survey.

Click here for additional data file.

Supplemental Information 3 Proportion of Landscape types within a research site1.

1 Bolded landscape types represent those consisting over 2% of the whole research area

Click here for additional data file.

Supplemental Information 4 Images of Landscape Type.

Click here for additional data file.

Supplemental Information 5 1st Survey Form.

Designed through 3 rounds of FGIs

Click here for additional data file.

Supplemental Information 6 2nd Survey Form.

Click here for additional data file.

Supplemental Information 7 Raw data derived from a First and Second Survey.

Click here for additional data file.

Supplemental Information 8 1st FGI Summary.

Click here for additional data file.

Supplemental Information 9 2nd FGI Interview Summary.

Click here for additional data file.

Supplemental Information 10 3rd FGI Interview Summary.

Click here for additional data file.

Additional Information and Declarations

Competing Interests

Author Contributions

Data Availability

The authors declare that they have no competing interests.

Baysok Jun conceived and designed the experiments, performed the experiments, analyzed the data, prepared figures and/or tables, authored or reviewed drafts of the paper, and approved the final draft.

Ilkwon Kim conceived and designed the experiments, performed the experiments, analyzed the data, prepared figures and/or tables, and approved the final draft.

Jihoon Shin conceived and designed the experiments, performed the experiments, authored or reviewed drafts of the paper, and approved the final draft.

Hyuksoo Kwon conceived and designed the experiments, authored or reviewed drafts of the paper, and approved the final draft.

The following information was supplied regarding data availability:

Raw data, survey results, summaries of three rounds of focus group interview are available in the Supplemental Files.

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
