# Peer review of "Development of landscape conservation value map of Jeju island, Korea for integrative landscape management and planning using conservation value of landscape typology"

_PeerJ, doi:10.7717/peerj.11449_

## Round 0.1 · original submission · Major Revisions

Dear Dr. Jun,

As you can see from the comments below, both qualified reviewers I have consulted acknowledge the originality and interest of your paper. At the same time, they raise a number of concerns primarily regarding (but not limited to) the structure of the paper and the validity of the findings, which seem to me quite major.

I kindly invite you to revise your manuscript following their indications and resubmit for further consideration. I will likely consult the Reviewers again for their opinion on the revised manuscript.

Kind regards,
Andrea Ghermandi (PeerJ editor)

Reviewer 1 ·

Basic reporting

The study addresses the interesting question of how to evaluate landscapes for application in conservation planning and management. It provides an interesting case study and demonstrates how land-cover and landform data can be combined with expert evaluations. The story is mostly easy to follow and rooted in relevant literature within the field. The paper's strength lies in the ambitious attempt to build a consensus on landscape values from a descriptive and largely value-neutral framework of landscape type classification. I enjoyed reading the article. I had some issues with the structure of the text and justification for some of the methodology and the findings.
I had some problems with the structure of the article. It mostly follows the IMRAD-structure of scientific writing (introduction; materials and methods; results; discussion), but several deviations from this structure occur. I would recommend restructuring some of the material to avoid mixing results with methods and methods with discussion. This applies to the sections mentioned below, but the authors should go through the review systematically to check for inconsistencies in this regard.
Line 233–244, line 287–296, and line 308–318 summarise land cover and the cover of landforms and landscape types in the study area. Perhaps these sections could be moved to the results chapter, or alternatively to the description of the study area. Anyway, these summaries are not methods.
I would recommend starting the discussion chapter by building on the results. Line 434–445 is a somewhat confusing start of the discussion and could be moved or omitted. I would suggest starting the discussion with line 446.
Line 463–468 is a description of methods and should be moved to the methods section.
The section about limitations and shortcomings (line 562–572) is relevant and exciting, but should be moved to the discussion section, whereas the conclusion section could be shorter. I recommend that limitations and shortcoming is included as a separate section in chapter 4.

The tables, figures and maps are excellent, and they convey the information in an effective manner. An exception is Figure 3, whic is distorted in some way (stretched), and should be redesigned.

Experimental design

The introduction provides an exciting stage of the paper. It explains the dilemmas and conflicts in landscape management in the country, i.e., the different and often irreconcilable views on future development/trajectories for a landscape. Line 141–145 gives some good clues, in this respect.
The justification for the use of land-cover maps and landform maps is well justified in theory and well explained. However, more references should be added to provide examples of the value criteria used for landscape evaluation (e.g, the value criteria shown in Figure 7 part 2 (representativeness, specificity, etc.; see e.g, Solecka, I. 2018. The use of landscape value assessment in spatial planning and sustainable land management — a review. Landscape Research, 1-16. doi:10.1080/01426397.2018.1520206 or Erikstad, L., Lindblom, I., Jerpåsen, G., Hanssen, M.A., Bekkby, T., Stabbetorp, O. & Bakkestuen, V. 2008. Environmental value assessment in a multidisciplinary EIA setting. Environmental Impact Assessment Review, 28(2-3), 131-143. doi:10.1016/j.eiar.2007.03.005).
Several key concepts and terms are not clearly defined, and many of the terms are often used in very different meanings and settings in landscape research. Therefore, I would recommend adding a box, including key definitions of terms such as landscape, landscape type, landscape value, and the value criteria in box 7 par 2.
It is not entirely clear to me whether the landform map was created by Park 2004, or only the method. Please try to write this in a more pointed manner. If I understood the manuscript (line 263–296) you did this based on the methodology from Park 2004, adapted to the scale of the study area. Please write this section shorter and somewhat clearer.

Validity of the findings

The process with multiple assessments in a focus group, including documentation of the process in the supplementary material, is clearly described and easy to follow. However, the connections between sub-criteria such as representativeness, rarity, etc., and the numbers that the experts used in the evaluation is not fully transparent. This should be elaborated and rooted in theory. General problems with expert judgment should also be discussed in more detail. I would argue that putting numbers on values is a good idea only as far as the criteria for values such as "high value", "medium value", etc. is clearly described and defined. The lack of broader user participation except from expert should also be discussed further (this is briefly mentioned in the conclusion section).
I also think the difference between general landscape types (reoccurring units) and individual landscape areas should be discussed more. The conceptual idea of assigning landscapes to types is rooted in the tradition of systematic physical geography or 'landscape geography', the aim of which is to present and explain typologies of similar landscapes based on their material content. Grouping of similar landscapes into types is an effective way to communicate landscape information because affiliation to type alone will provide an extensive amount of information about any singular individual of that particular type. Since a type in a type system comprises a predictable, 'normal' amount of landscape variation, affiliation to type is also a useful reference and a good starting point for assessment of the unique character and properties of individual landscapes. However, variation within types and individual characteristics of singular areas is not discussed in detail, although the issue is raised by some experts in the supplementary material. I think this should be discussed further.

Additional comments

With these taken into consideration, I am happy to recommend the manuscript for a new review and possibly publication.

Reviewer 2 ·

Basic reporting

This is an interesting study and the authors have developed the landscape conservation value for the Jeju Island using GIS and focus group interviews (FGIs). English is good but the paper should be well structured, especially in the introduction and discussion sections.

Experimental design

The article meet the standards.

Validity of the findings

I think all underlying data have been sufficiently provided,

Additional comments

As the paper currently stands, I have some questions and suggestions for making the paper more systematic.

Introduction:
It is necessary to restructure the introduction. It may contain some general content about Korea but too much local. I would recommend that the authors rewrite the paragraphs to be more concise and concrete in terms of presenting the core issues, research gap, and main questions.
In this regard, the front and back paragraphs should be connected, well structured. For example, the paragraph that starts with landform (i.e. Line 145) do not connect well with the preceding paragraph.

Materials and Methods:

2 of the Unit ‘km2’ should be superscripted.

What program was used to obtain the upslope contributing area (AS) and surface curvature (Cs)?

Line 288: spacing error, distinction(Fig. 4)

Where is Table B1?

Line 364: Relevant references should be added.
Line 372: is to have -> was to have

Results:

Line 413: In Figure 8 -> Figure 9?
Need to revise the legend of Figure 9 so readers can more easily check the high and low levels instead of quantile.

Table 3: In my understanding, the final LCV is standardized from 1 to 10. How do you get mean pixel LCV and Std. Deviation greater than 10? Please explain if there is anything I missed.

Discussion:
The discussion should be concise. For example, the starting paragraph is too common. In addition, it's a little unnatural with methodology and new results (i.e. susceptible areas).

Although the authors mentioned landscape terminology in terms of socio-ecological systems, human related cultural values will be omitted as the developed LCV deals with spatial content (i.e. land cover and landform).

Conclusion:
Line 553: In this context, land cover data…
That's what you might think, it is considered an extended interpretation that the land cover itself reflects cultural things.

---

## Round 0.2 · Minor Revisions

Dear Dr. Jun,

I have now received the feedback from the two reviewers that had reviewed your original submission. As you can see below, they are both satisfied with the changes you have made in the revised version, I congratulate you for that.

Before I follow through on their recommendation and accept the paper for publication though, I want to ask for some final minor adjustments.

It was brought to my attention that there are some privacy-related concerns in the current manuscript and supplementary information due to the inclusion of the names of the experts you have consulted and their direct quotes.

To avoid such potential issues, I recommend that you will remove the names of the experts altogether. Also, the usual policy of the journal for the inclusion of direct quotes is to recommend adhering to the COREQ guidelines and provide a thematic analysis of the quotes. I kindly ask you thus to either revise the reporting of the qualitative aspects (following the COREQ guidelines and providing a thematic analysis of quotes) or drop the direct quotes altogether, which do not seem essential to the manuscript.

On a final note, both reviewers noted that there are still some typos in the manuscript, so I suggest that you proofread the manuscript one more time before resubmitting it.

Congratulations on this interesting study and kind regards,
Andrea Ghermandi

Reviewer 1 ·

Basic reporting

The issues with structure and clarity from the first submission is solved, and the revised manuscript is much clearer and more readable. The english is good, but some references appears to be at the beginnng of a sentence without furter introduction (at least in my tracked-changes-version). I have not proofread the manuscript, so pleas ensure high-quality proof-reading befor publication.

Experimental design

Comments from the first review are incorporated nicely. Problematic issues with the experimental design is now solved.

Validity of the findings

Comments from the first review are incorporated nicely, with a much better discussion of limitations.

Additional comments

Thank you for incorporating the comments from the peer-review in a wise and elegant manner. I think the manuscript is much improved, and shoul be ready for publication after proofreading.

Reviewer 2 ·

Basic reporting

Overall, the manuscript has been improved according to the comments. But, the typos are still found (e.g. Line 116: n terms of). The authors are advised to correct the ms thoroughly.

Experimental design

No comment

Validity of the findings

No comment

Additional comments

Thank you for your hard work.

---

## Round 0.3 · accepted · Accept

Thank you for revising the manuscript. Congratulations for the interesting work!